# 3D-Printed Soft Pneumatic Robotic Digit Based on Parametric Kinematic Model for Finger Action Mimicking

**DOI:** 10.3390/polym14142786

**Published:** 2022-07-07

**Authors:** Shumi Zhao, Ziwen Wang, Yisong Lei, Jie Zhang, Yuyao Li, Zeji Sun, Zidan Gong

**Affiliations:** 1Sino-German College of Intelligent Manufacturing, Shenzhen Technology University, Shenzhen 518118, China; 2070412024@email.szu.edu.cn (Z.W.); leiyisong2020@email.szu.edu.cn (Y.L.); 2110412004@stumail.sztu.edu.cn (J.Z.); arizalyy@foxmail.com (Y.L.); sunzeji0322@foxmail.com (Z.S.); 2Institute of Artificial Intelligence, Hefei Comprehensive National Science Center, Hefei 230026, China; zhaoshumi@iai.ustc.edu.cn

**Keywords:** soft robotic digit, physical properties, mechanical properties, analytical modeling

## Abstract

A robotic digit with shape modulation, allowing personalized and adaptable finger motions, can be used to restore finger functions after finger trauma or neurological impairment. A soft pneumatic robotic digit consisting of pneumatic bellows actuators as biomimetic artificial joints is proposed in this study to achieve specific finger motions. A parametric kinematic model is employed to describe the tip motion trajectory of the soft pneumatic robotic digit and guide the actuator parameter design (i.e., the pressure supply, actuator material properties, and structure requirements of the adopted pneumatic bellows actuators). The direct 3D printing technique is adopted in the fabrication process of the soft pneumatic robotic digit using the smart material of thermoplastic polyurethane. Each digit joint achieves different ranges of motion (ROM; bending angles of distal, proximal, and metacarpal joint are 107°, 101°, and 97°, respectively) under a low pressure of 30 kPa, which are consistent with the functional ROM of a human finger for performing daily activities. Theoretical model analysis and experiment tests are performed to validate the effectiveness of the digit parametric kinematic model, thereby providing evidence-based technical parameters for the precise control of dynamic pressure dosages to achieve the required motions.

## 1. Introduction

Wearable soft robotic gloves with basic hand restore functions were developed to improve the quality of life of individuals with neurological impairment or who have experienced hand trauma [1,2,3]. Soft robotic actuators made from linear or nonlinear soft materials [4,5,6] are responsible for the predesigned force or motion in different terminal devices [7,8,9]. Such components are lightweight, flexible, and compatible with human-machine interactions and are widely utilized in the fields of medical surgery and rehabilitation [10,11]. Soft actuators with chambers can offer smooth and flexible bending motions, making them ideal to be applied in soft robotic gloves for hand rehabilitation [12,13]. The bending bellows actuator is a typical long beam-like element, of which one side is longer than the other along the longitudinal direction, resulting in actuator bending [14]. To achieve differential stretching, the bellows are partitioned into several compartments before different pressure supplies are provided to each chamber, thereby achieving different bending angles [15]. Rotary flexible actuators, which are typically used in two robotic links, can rotate a particular geometry to provide movement [16]. Existing soft actuators designed for finger exoskeletons adopt a single-structure actuator [17,18,19]. A single-structure actuator for a finger is easy to manufacture, but hand motion freedom is limited by the single-direction movement [20,21]. With the development of soft robotic technology, traditional rigid mechanical and single-structure designs transitioned gradually to soft actuated and bionic finger musculoskeletal structure designs.

Thermoplastic polyurethane (TPU) is a new environmentally friendly linear block copolymer material composed of soft and hard segments, which can be processed using common thermoplastic processing methods [22,23]. In the processing, diisocyanates and small molecule chain extenders (e.g., diamines or diols) create the hard segments of TPU, providing satisfactory mechanical strength properties, whereas oligomeric diols generate the soft segments of TPU, realizing flexibility and elastic behavior [24,25]. Thus, depending on the mixed ratio of soft and hard segments, the material properties of TPU (e.g., structural morphologies, high tensile strength, satisfactory chemical resistance, and machinability) can be customized to demonstrate different performance levels that apply in daily used objects, sporting goods, toys, and decorative materials [26,27]. Furthermore, TPU products can be generated using 3D printing technology, which is also known as additive manufacturing, to construct objects by printing individual layers based on a digital model file and guarantee the manufacturing of complex structures [28,29]. Compared with traditional processes, 3D printing can directly and easily fabricate products with complex structures without the use of molds [29,30]. Currently, two fabrication methods exist for 3D printing using TPU: fused deposition modeling (FDM) and selective laser sintering [31,32]. FDM is the most widely used 3D printing method owing to its simple operation, mature technology, relatively cheap equipment investment, and compatibility with a wide range of materials [33,34]. Therefore, the use of FDM-based 3D printing to fabricate pneumatic actuators could be an ideal choice due to its advantages [35].

The bending angle and motion trajectory of a robotic digit produced by soft actuators are typically measured via experiments or analyzed using mathematical and finite element models. However, constructing an accurate model is difficult owing to the highly nonlinear characteristics of the materials used [36,37] and complex coupling between human fingers and actuators [18,38]. Previous studies have suggested that numerical functions could describe nearly any mechanical object, and the finite element method is able to simulate the behavior of pneumatic bellows actuators [39,40]. Polygerinos et al. [41] used to present a quasistatic model for a soft fiber-reinforced actuator to analyze the bending angle, as well as the contacting force of the object. Based on elastica theory by Euler, Payrebrune et al. [42] developed a rod-based model to predict the bending deformation and tip force of a PneuNets soft actuator when several parameters are determined before the experiments. However, for a soft pneumatic robotic finger, such methods require precise object description and complex computations [43,44]. To understand the behavior of soft actuators, especially 3D-printed hyperelastic material (e.g., TPU [45]) actuators, the stress-strain relationship of TPU should be determined via an experiment test before the actuator design, and finite element simulations of soft actuators based on such material parameters can predict their quasistatic behavior [46,47]. Therefore, modeling is necessary to develop a soft robotic finger, as it can help generate geometrical design parameters and predict the bending angle and grasping force of actuators in the control process, thereby reducing costs and experimental exploration time [38,42,48]. However, different materials have different properties, and pneumatic actuators with a simple structure can easily bend into an arc, which is inconsistent with the curved contour of fingers. Therefore, developing an actuator that can mimic a human finger is difficult for designers [38,49]. To create a bending outline for pneumatic actuators that conforms to the shape of a finger, Yang et al. [50] attached heterogeneous components to the bottom part of an actuator. However, changes in the number or distribution of bellows chambers based on the anatomy of fingers have yet to be presented. Thus, investigating the process of matching the bending outline of pneumatic bellows to the curved contour of fingers is necessary.

The present study aims to design and develop a 3D-printed soft pneumatic robotic digit based on a parametric kinematic model. In addition, actuator parameters such as the pressure supply, material requirements of the bellows actuators, and tip trajectories are analyzed and determined. The soft pneumatic robotic digit is constructed by adopting three connected pneumatic bellows actuators, which are designed as flexible chambers to achieve specific finger motion trajectories such as bending and extending, under different pneumatic pressure supplies. The direct 3D printing technique is used in the fabrication process of the soft pneumatic robotic digit, and the mechanical properties of the pneumatic bellows actuators are validated via experiments. The motion trajectories of the robotic digit and the motion range of each joint are described and measured based on the parametric kinematic model.

## 2. Methods and Device Fabrication

### 2.1. Soft Pneumatic Robotic Digit Design

Given the complexity of robotic finger motions, understanding the natural skeletal structure of a human finger is essential (Figure 1a,b). A finger mainly contains three interphalangeal joints [49,51], namely, a distal (DIP), proximal (PIP), and metacarpal (MCP) joint. According to previous studies [52], the range of motion (ROM) of finger joints is approximately the same in different individuals. The bending angles of MCP, PIP and DIP joints are *φ*1, *φ*2 and *φ*3 respectively. Activation of the extrinsic and intrinsic muscles controls finger movements [53], and the maximum flexion value of the MCP, PIP, and DIP joint ranges is approximately 95°, 110°, and 90°, respectively [54].

Based on the natural skeletal structure of a human finger, a compliant soft robotic digit with pneumatic bellows actuators is designed to mimic finger activities, as demonstrated in Figure 1c. Three pneumatic bellows actuators as biomimetic artificial joints are connected by the anthropomorphic bones to form a soft robotic digit, thereby realizing bending motion, with the selective actuation of each soft section for precise control. Figure 1d,e displays the designed soft pneumatic robotic digit using pneumatic bellows actuators and inner structure. The bellows chambers of the DIP, PIP, and MCP joints of the finger can be inflated and deflated with air pressure to generate different bending angles. The parameters of the pneumatic bellows actuators, such as length and number, are determined based on the motion distance of fingers and other factors.

### 2.2. Actuator Model Mimicking Finger Joints

The developed soft pneumatic robotic digit consists of a certain number of pneumatic bellows actuators, with a geometry allowing flexion and extension motions during air pressure inflation and deflation, respectively. The pneumatic bellows actuator has several half circle chamber segments arranged on the base, as shown in Figure 2a,b. The initial length of pneumatic bellow actuator is *L*_o_ and its one convolution is *a*_1_. *R*_1_, *R*_2_ represent the radius of small and large chamber respectively. When inflated air pressure (*P*) exists in the actuator chambers, the geometry changes and bending motion occurs in the y direction.

The bending angle *φ* is realized by the combined effect of torque, which is generated by the internal pressure *P* at the distal cap of the actuator, and the differential expansion between the top and base parts of the actuator generating a bending radius (*R*), as shown in Figure 2b,c. Therefore, the actuator realizes a specific bending angle (*φ*) at a given actuation pressure (*P*), which can be given by a simplified mathematical model by assuming that all the interacting forces (including the external force (*F_e_*)) are balanced in a quasistatic condition. A moment equilibrium around point O (Figure 2c) can be delivered to the net torque formulation, as follows by Equation (1):(1)$$M_{net}=M_{P}-M_{r}-M_{F}$$
where *M_p_* is the summation of the actuation torque caused by the internal pressure, *M_r_* is the resistive torque owing to the resistance of the pneumatic bellows against the deformation, and *M_F_* is the external torque resulting from the external force (*F_e_*) exerted at the end of the actuator. In the equilibrium configuration shown in Figure 2b,c, the net torque (*M_net_*) is zero. The length of the pneumatic bellows actuator base (*L_0_*) is assumed to be constant even after bending, where Δ*L* is the deformation length of the pneumatic bellows actuator, *a*_1_ is the convolution length of bellows, and *N* is the number of bellows.

The torque *M_p_* generated by the inflated air pressure is calculated as follows by Equation (2):(2)$$M_{P}=P\frac{\pi }{2}r_{1}^{2}(\frac{4}{3\pi }r_{1}+b)$$

Equation (2) describes the linear relationship between the generated torque and inflated pressure when the geometric features of the actuator are determined. The geometric parameters of the pneumatic bellows actuator in Equation (2) are defined in Figure 2c,d. Bottom thickness of the actuator is *b*, chamber wall thickness of TPU is *t*, internal radius of small chamber is *r*_2_, and internal radius of large chamber is *r*_1_.

The pneumatic bellows actuators are made from TPU 801, and the nonlinear mechanical properties of such material could be examined using nonlinear elasticity theory [55,56]. The Yeoh third-order model is an available model that is suitable for the nonlinear behavior characterization of materials, despite its elongation percentage being above 300% [55,57]. The Yeoh stress-strain energy function of an isotropic material can be determined by Equation (3):(3)$$W=\sum_{i=1}^{3}C_{i}{\left(I_{1}-3\right)}^{i}$$
where *C_i_* represents the coefficients of the polynomial function, which can be calculated via experiment test, and *I*_1_ is the first strain invariant based on the principal stretches (*λ_i_*), as followed Equation (4):(4)$$I_{1}={\sum_{i=1}^{3}\lambda_{i}}^{2}$$

Assuming that the condition of the TPU material is incompressible, the relation of the principal stretches (*λ_i_*) is as followed Equation (5):(5)$$\lambda_{1}\lambda_{2}\lambda_{3}=1$$

Along the axial direction of the actuator, the principal stretches (*λ*_1_) are defined as *λ*_1_ = *λ*; thus, the other principal stretches (*λ*_2_ and *λ*_3_) along the radial and circumferential directions can be derived from Equation (5) as Equation (6):(6)$$\lambda_{2}=\lambda_{3}=\frac{1}{\sqrt{\lambda }}$$
where *λ*_2_ is equal to *λ*_3_, assuming that uniaxial stress is applied dominantly in the axial direction. Therefore, substituting the principal stretches in Equation (4) can yield Equation (7):(7)$$I_{1}=\lambda^{2}+\frac{2}{\lambda }$$

For the pure uniform strain, the uniaxial stress is calculated using the energy function in Equation (3) based on the invariant strain *I_i_* (I = 1,2,3 corresponding to the first, second, and third strain invariant, respectively), as followed by Equation (8) [58,59]:(8)$$\sigma_{yeoh}=2(\lambda^{2}-\frac{1}{\lambda })(\frac{\partial W}{\partial I_{1}}+\frac{1}{\lambda }\frac{\partial W}{\partial I_{2}})$$

Substituting Equation (3) in Equation (8) yields Equation (9):(9)$$\sigma_{yeoh}=2(\lambda^{2}-\frac{1}{\lambda })(C_{1}+2C_{2}(I_{1}-3)+3C_{3}{\left(I_{1}-3\right)}^{2})$$

To calculate the stress, the cross section of the soft pneumatic bellows actuator (Figure 1d) is split into two zones, namely, the base and arc sections. The stress of each zone is calculated using Equation (9), as followed Equations (10) and (11):(10)$$\sigma_{b}=2(\lambda_{b}{}^{2}-\frac{1}{\lambda_{b}})(C_{1}+2C_{2}(\lambda_{b}{}^{2}+\frac{2}{\lambda_{b}}-3)+3C_{3}(\lambda_{b}{}^{2}+\frac{2}{\lambda_{b}}-3{)}^{2})$$
(11)$$\sigma_{t}=2(\lambda_{t}{}^{2}-\frac{1}{\lambda_{t}})(C_{1}+2C_{2}(\lambda_{t}{}^{2}+\frac{2}{\lambda_{t}}-3)+3C_{3}(\lambda_{t}{}^{2}+\frac{2}{\lambda_{t}}-3{)}^{2})$$
where *λ_b_* and *λ_t_* are the strain of the base and arc sections, respectively.

For the material properties of the pneumatic bellows actuators against the deformation caused by pressure, any points (red point as shown Figure 2d, its position in y direction is *y*, *θ* is the angle and *r* is radius of the red point in the cross section project of the pneumatic bellows actuator) of the pneumatic bellows actuators would generate a resistance force, and therefore, the resistive torque is derived as followed Equation (12):(12)$$M_{r}=\int \sigma_{b}ydA_{b}+\int \int \sigma_{t}(b+r\sin\theta )dA_{a}$$
where *dA_b_* and *dA_a_* represent the small strain area units of the base and arc sections, respectively. Equation (12) does not have a closed form analytical solution; thus, it is computed numerically, and the curves fit.

For the actuators of the soft robotic digit, two different cases are examined. The first case involves no external force, that is, free-end motion, and the second case involves an external force, that is, constrained-end motion. Equation (1) yields the relationship between the bending angle (*φ*) and a function of the internal pressure (*P*), assuming the equilibrium point, as followed Equation (13):(13)ϕ=f(P)

When an external force acts on the actuator, that is, constrained-end motion, the external torque at the equilibrium point can be determined as Equation (14):(14)$$M_{F}=M_{P}-M_{r}$$

Therefore, the external torque at the equilibrium point can be used a function *g (φ, P)* which depends on the inflated pressure (*P*) and the bending angle (*φ*) and could be expressed through a nonlinear Equation (15).
(15)$$M_{F}=g(\phi ,P)$$

When the pneumatic bellows actuators are inflated with air mass, pressure increases as well as the *φ*, and the pressure-controlled motion trajectory of the pneumatic bellows actuators can be determined. Thus, the simplified mathematical model can be used to predict the motion of the pneumatic bellows actuators from a quantitative or qualitative perspective.

### 2.3. The 3D Printing Fabrication

Direct 3D printing in the fabrication of soft robots is a new technique without requiring an additional casting process [60,61]. The fabrication of the 3D-printed soft robotic digit involves three steps. First, the appropriate materials (e.g., TPU) that can realize the desired characteristics of the final product are meticulously selected. Second, the printing method is determined based on the material properties, structure size, and vertical movement of the nozzle and thickness of the layers. TPU materials can be processed into a filament and are compatible with the FDM process to fabricate the complex structure of the soft actuator [62]. Finally, the print file (i.e., soft pneumatic robotic digit model) and print material are loaded into the 3D printer, and printing begins. In addition, the primary 3D printing procedure starts with a computer-aided design (CAD), in which the print file (i.e., soft pneumatic robotic digit model converts to stereolithography (STL) format) is envisaged and drawn with CAD software [63]. Next, the 3D soft pneumatic robotic digit model is divided into multiple 2D stack layers piled up via a printer jet nozzle depending on the precision of the 3D printer. The bellows chamber is filled with water-soluble material. Subsequently, the TPU model is cured, and the water-soluble material is cleaned using water. The fabrication process of the soft robotic digit is shown in Figure 3a.

The soft pneumatic robotic digit made from TPU presented in Figure 3b consists of three pneumatic bellows actuators as biomimetic artificial joints connected by anthropomorphic bones. Considering the ROM of fingers, the kinematic movements of biological fingers should be replicated by the soft actuators as much as possible to customize each bending bellows actuator corresponding to the MCP, PIP, and DIP joints by changing the length of the phalanges. Such actuators can be used to match the kinematics of a human finger to mimic various finger activities.

## 3. Results and Discussion

### 3.1. Pneumatic Bellows Actuator Analysis

As mentioned in Section 2.1, a simplified static mathematical model for the pneumatic bellows actuators only involves a small set of calculations based on several assumptions, and the bending angle is relative to the internal pressure of the chamber in free-end motion. The unknown parameters (i.e., C_1_, C_2_, and C_3_) of the TPU material used for the model calculation are determined by a curve fitting [64], as shown in Figure 4, to obtain the values presented in Table 1. In the case of small deformation, C_1_ represents the initial shear modulus, which is positively related to the stiffness of the material. The second coefficient C_2_ is negative, which can reflect the softening of the material at moderate deformation. However, the third coefficient, namely, C_3_, is positive, which can describe the hardening of the material in the case of large deformation.

Table 2 lists the values used in the soft actuator model for the Equation (13) calculation. Equation (12) can be solved by employing the definite integral and multiple integral methods in MATLAB (MATLAB 2014, MathWorks, Natick, MA, USA) [65], based on the physical parameters reported in Table 2. The model shows that a pneumatic bellows actuator can work under low pressure, owing to the nonlinear mechanical properties of the TPU material. The model demonstrates satisfactory agreement with the experiment results as the pressure increases. Therefore, the proposed model can be adopted for the finger unit design.

Figure 5a–f presents the interactions between the vital parameters of multi-material pneumatic actuators revealed by the MATLAB simulation. To achieve the same bending angles, the required pressure would decrease as the *N* and the radius of the large bellows chambers increase (Figure 5a,b). Through different bending angle tests (e.g., angle = 120°, 90°, and 60°), while maintaining the same radius of the large chamber, the bending angle of the bellows increases along with the increase in pressure. Conversely, the bending angle of the bellows increases slowly as the small chamber radius increases, as shown in Figure 5c. The increased angle caused by changing the large radius is larger than that caused by the small radius. Moreover, Figure 5d,e show that, when the thickness of the bottom substrate increases, the pressure would increase to sustain the same bending angle and the bending angle of the bellows wall decreases at the same pressure; whereas when the thickness of the top bellows increases, the bending angle variation trend is similar. In addition, the bending angle of the bellows increases as the stiffness of the actuator increases, because the stiffness is positive in the shear modulus C1 of the material properties, as shown in Figure 5f. The increase in the shear modulus of the actuator has obvious pressure requirements. Thus, the effects of multiple factors (e.g., length of bellows and chamber radius and thickness) on the multi-material pneumatic actuators can be analyzed quantitatively and qualitatively based on the simulated results.

The material and structure parameters can be utilized to tailor the bending angle of the bellows actuators and design the robotic digit in accordance with the requirements of specific finger applications. This study finds that the key parameters (e.g., the thickness of the bellows and radius and structure of the chambers) and pressure supply directly affect the bending angle of pneumatic bellows actuators.

### 3.2. Parametric Kinematic Model Analysis for the Robotic Digit Motion

Following finger structure flexion and extension, a simplified robotic digit configuration is adopted, as shown in Figure 6. To support finger flexion and extension comfortably, the three biomimetic artificial joints fabricated using pneumatic bellows actuators should follow the ROM of a human finger and generate adequate force to bend the finger properly. A parametric kinematic model of the robotic digit is presented from the kinematic perspective of the pneumatic bellows actuators, assuming no impact from artificial bones is exerted. References are set based on the finger frames and assigned to the model to depict the bending action of joint points and tip point C in Figure 6a. The position variation of the tip point in the bending direction can be described using its components by the newly designed kinematic model and compared with the traditional kinematic model [66], as shown in Table 3.

In the newly designed model, *φ*_1_, *φ*_2_, and *φ*_3_ are the first, second, and third actuator joint bending angles, respectively; *L*_1_, *L*_2_, and *L*_3_ represent the length of each biomimetic artificial joint; and *L*_11_, *L*_12_, and *L*_13_ represent the length of the phalanges. Figure 6a indicates the configuration of the inertial frame and relative angles between the adjacent rigid sections at each joint. The finger bends when the robotic digit is inflated. Figure 6b,c show the locations of the reference (0,0) and tip (*x*,*y*) points and the robotic digit working in the bending stage, with the digit tip moving trajectory specified by a yellow marker in the traditional kinematic model and newly developed kinematic model, respectively. The trajectory of the digit tip is simulated as shown in Figure 6d. Under low pressure, the *x* position of the digit tip has a large gap, and under high pressure, the y position of the digit has a large gap between the two kinematic models. In the traditional kinematic model, the start reference point is not static and can be changed when pressure is provided in the digit. In the newly developed kinematic model, the start reference point is static and cannot be changed when the soft pneumatic digit is moving, which can precisely achieve the digit tip moving trajectory.

### 3.3. Robotic Digit Motion Analysis

The trajectory of the digit tip and ROM of the DIP actuator (the first actuator) are simulated and validated in the soft pneumatic robotic digit experiments, in which different points are tested under various constant pressure supplies, as shown in Figure 7. When the air pressure in the bellows chamber increases gradually from 0 kPa to 30 kPa, the joint ROM of the first actuator is within the range of 0–97°. Figure 7a displays the simulation and experiment results towards the angular displacement variation of the first bellows actuator with respect to the air pressure inflation dosage. The difference in the bending characteristics of the pneumatic bellows actuator under different pressure supplies between the simulation and experiment is less than 13%, as shown in Figure 7b. The ROM of the different bellows actuators is highly dependent on the pressure supply. The results suggest that the bellows actuators model and experiment are consistent and the bellows actuator can be tailored for different link joints. Therefore, the MCP, PIP, and DIP joints are fixed with different chamber-structure bellows actuator numbers, that is, 7, 5, and 4, respectively. Figure 7c,d presents the comparison of the robotic digit tip index trajectories in both the experiment (as red point in Figure 7e–g) and kinematic model simulation. The two trajectory path plots behave similarly and exhibit similar inflection points, but the experimental data outcomes, especially at high pressure, reflect the new model more closely than the traditional one, which suggests that the new kinematic modeling improves the prediction accuracy of the moving trajectory because the start reference point is static and does not change when the soft pneumatic digit moves. Therefore, based on the established kinematic model, the soft actuators can be customized for different applications by changing their material and geometric features.

The measured ROM of the soft robotic digit satisfies the finger bending range and is close to the reported anatomical ROM and simulated results in Figure 7. The slight variations between the experiment and simulation results can be attributed to the weight effect of the digit and friction between the fabricated robotic digit and flat supporting surface during flexion and extension. Therefore, an actuation pressure of over 30 kPa is required to provide the robotic digit with full motion range. Moreover, the achieved ROM (MCP = 107°, PIP = 101°, and DIP = 97°) is consistent with the functional ROM of a human finger for performing daily activities [67,68].

### 3.4. Robotic Digit Mimicking Finger Actions

Pneumatic bellows actuators can be tailored as biomimetic artificial joints according to finger sizes. The direct 3D printing technique is employed to fabricate a soft pneumatic robotic digit without requiring an additional casting process. The soft robotic digit can be controlled to bend under different pressure supplies. In the soft robotic digit motion range experiments, the MCP, DIP, and PIP joints bend at suitable angles, mimicking finger actions, as shown in Figure 8a–f. These results reveal that the developed soft robotic digit has the potential to match the bending changes in a human finger structure. 

The bending experiments on the basic movements of the soft robotic digit and gripping experiments for different objects (i.e., an empty beaker, empty square bottle, and lightweight cylinder), as presented in Figure 8g–i, demonstrated that the soft robotic digit can mimic finger actions. The different sizes of the objects can be grasped by the robotic digit through the extension and flexion mechanisms of the soft robotic digit, which can be easily controlled by pneumatic pressure. The object grasping experiment results show that the developed soft robotic digit can not only mimic finger flexion and extension, but also grip various types of objects for daily use. Although the grasp experiments are simple, the primary focus of this work is the development and optimization of a soft robotic digit based on a parametric kinematic model. In the future, the soft robotic finger will be evaluated for basic gripping motions to assist patients with finger movement difficulties.

## 4. Conclusions

This study proposed a soft pneumatic robotic digit with pneumatic bellows actuators using the 3D printing fabrication technique based on a parametric kinematic model, which matched human finger motions well. The parametric kinematic model was used for the pneumatic bellows actuator and finger motion analysis to predict the structure and material property parameters of the actuator design, such as the pressure supply and material requirements. In addition, the effectiveness of the proposed model was validated by the relevant experiment results. Pneumatic bellows actuators were designed as biomimetic artificial joints to satisfy the anatomical ROM of a finger. The geometry of the robotic digit allowed for forward and backward bending motions during inflation and deflation, respectively. Each digit joint achieved different motion ranges (MCP = 107°, PIP = 101°, and DIP = 97°) under a low pressure of 30 kPa, which are consistent with the functional ROM of a human finger. The experiments demonstrated the bending characteristics of the pneumatic bellows actuators under different pressure supplies. Theoretical and experimental analyses were also conducted to examine the mechanical properties of the soft pneumatic robotic digit. The performance of the soft pneumatic robotic digit was assessed via flexion and extension operations under various pressure supplies. The results showed that the soft pneumatic robotic digit can satisfactorily mimic human finger flexion and extension activities. In the future, studies on finger-impaired patient (e.g., stroke patients) assistance using the developed soft robotic digit could be conducted to evaluate the control strategies of the soft robotic digit system and further improve the soft robotic digit design to fit custom applications.

## Figures and Tables

**Figure 1 polymers-14-02786-f001:**
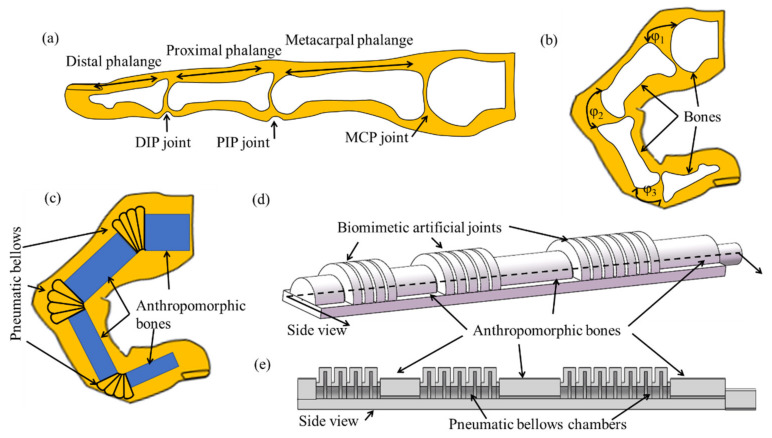
Finger structure and soft robotic digit structure: bone structure of a human finger in (**a**) extension and (**b**) flexion; (**c**) soft robotic digit mimicking finger flexion; (**d**) soft robotic digit with biomimetic artificial joints and (**e**) side view.

**Figure 2 polymers-14-02786-f002:**
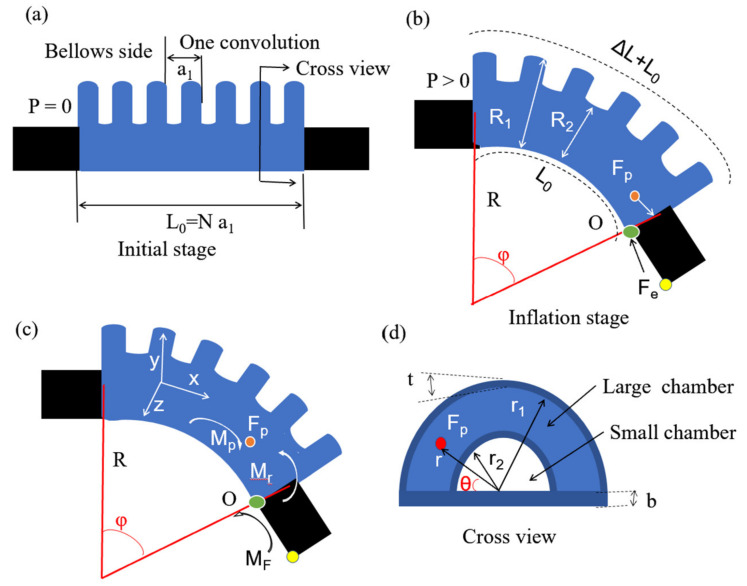
Pneumatic bellows actuator: (**a**) schematics of pneumatic bellows actuator between two connection parts; (**b**,**c**) bending analysis of the pneumatic bellows actuator structure; (**d**) cross section of the pneumatic bellows actuator.

**Figure 3 polymers-14-02786-f003:**
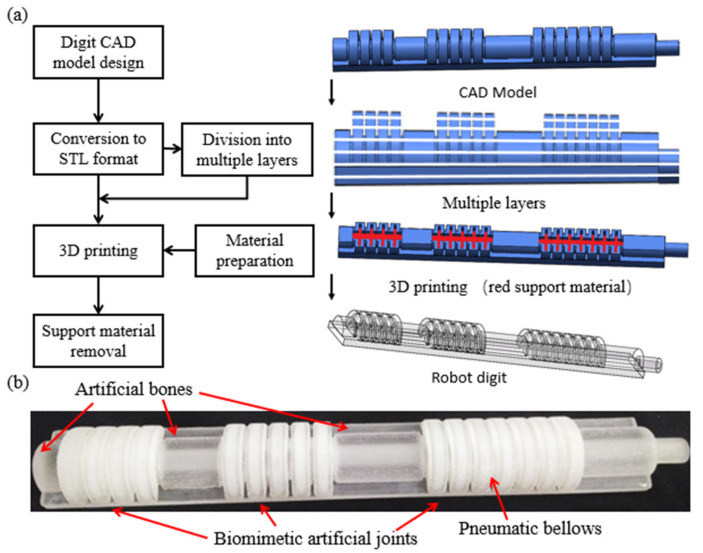
The 3D printing fabrication: (**a**) fabrication steps of 3D-printed soft actuators; (**b**) soft pneumatic robotic digit with biomimetic artificial joints.

**Figure 4 polymers-14-02786-f004:**
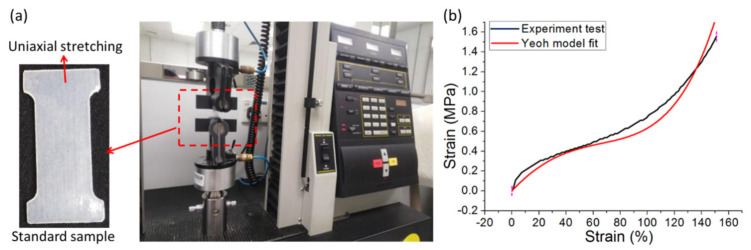
Parameter identification of Yeoh third-order model in uniaxial tension test: (**a**) the uniaxial tension test setup; (**b**) Yeoh model fit curve (red) and experiment curve (black).

**Figure 5 polymers-14-02786-f005:**
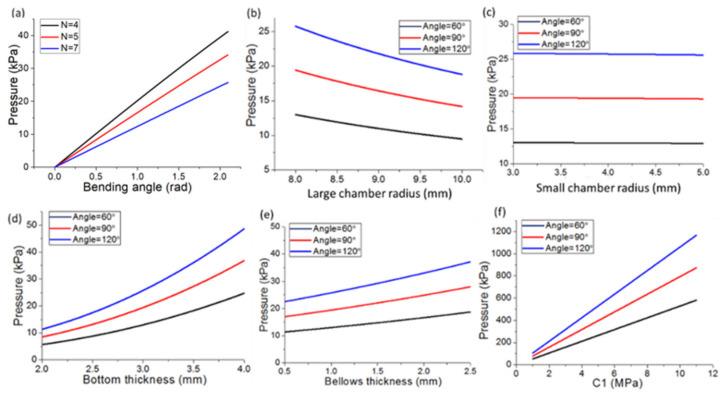
Analysis of the proposed bellows actuator model: (**a**) bellows number effects; (**b**) bending angle variation along with large chamber radius and (**c**) small chamber radius; (**d**) bottom substrate thickness; (**e**) bellows’ wall thickness; (**f**) actuator stiffness.

**Figure 6 polymers-14-02786-f006:**
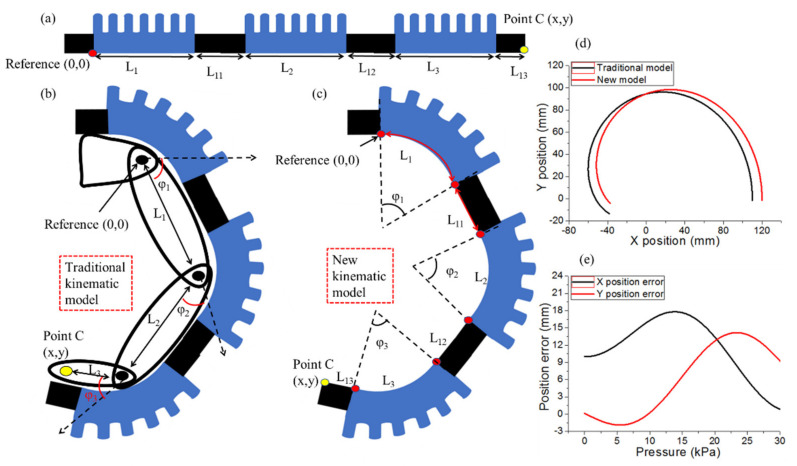
Comparison analysis of the proposed robot digit model and traditional model: (**a**) the initial stage; (**b**) traditional model bending and (**c**) new model bending stage (tip point marked by yellow dot); (**d**) tip trajectory comparison between two models; (**e**) position difference analysis.

**Figure 7 polymers-14-02786-f007:**
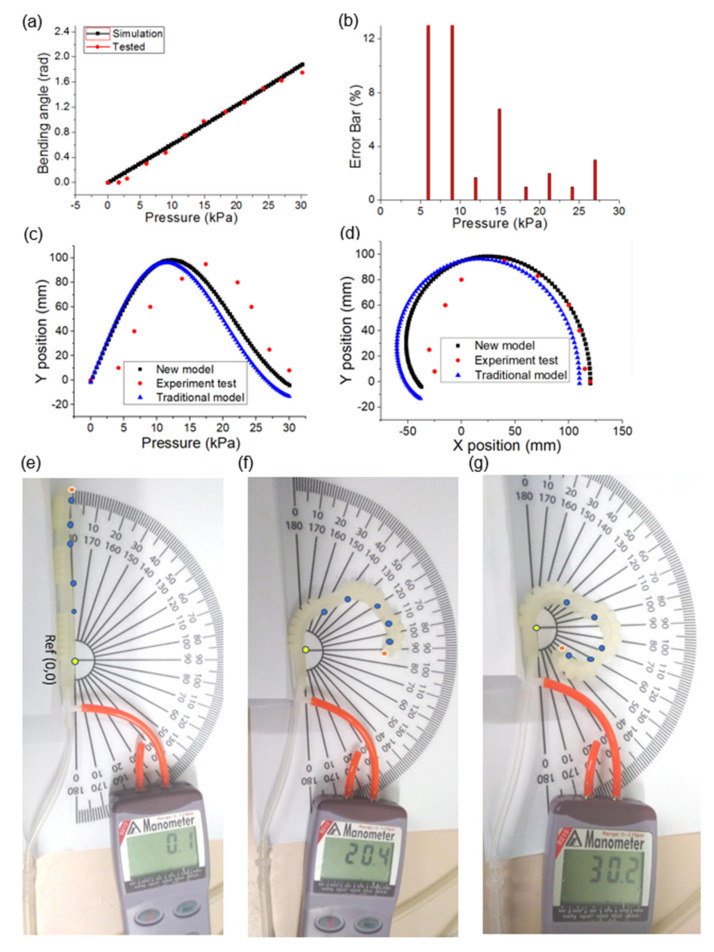
Single actuator and robotic digit analysis: (**a**,**b**) DIP actuator simulation and experiment comparison analysis; (**c**,**d**) schematic trajectory of soft robotic digit tip in experiment and kinematic model simulation; (**e**–**g**) experiment tests.

**Figure 8 polymers-14-02786-f008:**
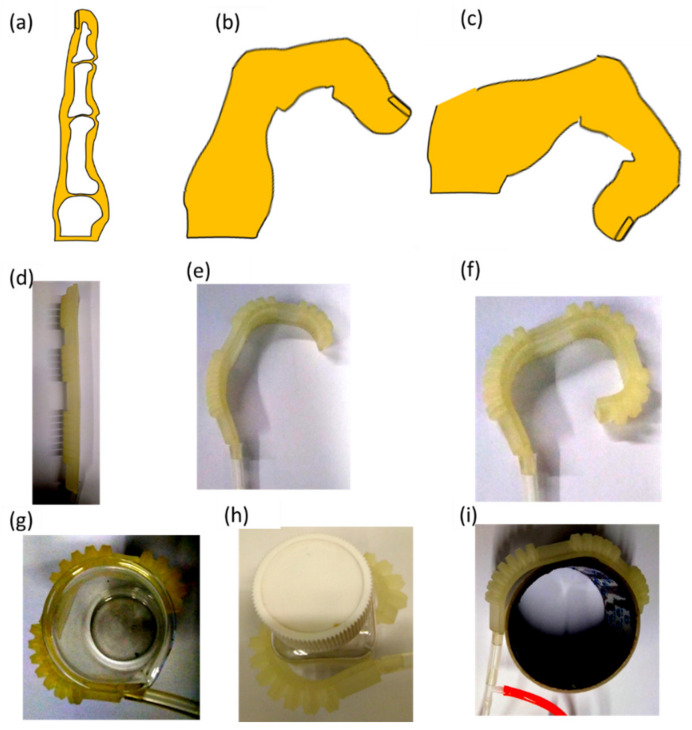
Soft robotic digit mimicking finger actions (**a**–**f**) and grasping three types of objects: (**g**) beaker; (**h**) square bottle; (**i**) cylinder.

**Table 1 polymers-14-02786-t001:** Coefficients of Yeoh third-order model.

Material	C_1_ (MPa)	C_2_ (MPa)	C_3_ (MPa)
TPU 801	0.2467	−0.0471	0.0102

**Table 2 polymers-14-02786-t002:** Structure parameters of pneumatic bellows actuators.

Parameters	Definition	Value
*b*	Bottom thickness of actuator	3 × 10^−3^ m
t	Chamber wall thickness of TPU	1 × 10^−3^ m
r_2_	Internal radius of small chamber	4 × 10^−3^ m
R_2_	Representative radius of small chamber	5 × 10^−3^ m
r_1_	Internal radius of large chamber	8 × 10^−3^ m
R_1_	Representative radius of large chamber	9 × 10^−3^ m
N	Number of bellows	7
a_1_	Length of one bellow	5 × 10^−3^ m

**Table 3 polymers-14-02786-t003:** Kinematic model comparison between traditional model and newly designed model.

Pressure	Traditional Kinematic Model	Newly Designed Kinematic Model
P = 0	$x=L_{1}+L_{2}+L_{3}$	$x=L_{1}+L_{11}+L_{2}+L_{12}+L_{3}+L_{13}$
	y=0	y=0
P > 0	$\begin{array}{l} x=L_{1}\cos\phi_{1}+\\ L_{2}\cos(\phi_{1}+\phi_{2})+\\ L_{3}\cos(\phi_{1}+\phi_{2}+\phi_{3}) \end{array}$	$\begin{array}{l} x=\frac{L_{1}}{\phi_{1}}\tan\frac{\phi_{1}}{2}+(\frac{L_{1}}{\phi_{1}}\tan\frac{\phi_{1}}{2}+L_{11}+\frac{L_{2}}{\phi_{2}}\tan\frac{\phi_{2}}{2})\cos\phi_{1}\\ +(L_{12}+\frac{L_{2}}{\phi_{2}}\tan\frac{\phi_{2}}{2}+\frac{L_{3}}{\phi_{3}}\tan\frac{\phi_{3}}{2})\cos(\phi_{1}+\phi_{2})\\ +(\frac{L_{3}}{\phi_{3}}\tan\frac{\phi_{3}}{2}+L_{13})\cos(\phi_{1}+\phi_{2}+\phi_{3}) \end{array}$
	$\begin{array}{l} y=L_{1}\sin\phi_{1}+\\ L_{2}\sin(\phi_{1}+\phi_{2})+\\ L_{3}\sin(\phi_{1}+\phi_{2}+\phi_{3}) \end{array}$	$\begin{array}{l} y=(L_{11}+\frac{L_{2}}{\phi_{2}}\tan\frac{\phi_{2}}{2}+\frac{L_{1}}{\phi_{1}}\tan\frac{\phi_{1}}{2})\sin\phi_{1}\\ +(L_{12}+\frac{L_{2}}{\phi_{2}}\tan\frac{\phi_{2}}{2}+\frac{L_{3}}{\phi_{3}}\tan\frac{\phi_{3}}{2})\sin(\phi_{1}+\phi_{2})\\ +(\frac{L_{3}}{\phi_{3}}\tan\frac{\phi_{3}}{2}+L_{13})\sin(\phi_{1}+\phi_{2}+\phi_{3}) \end{array}$

## Data Availability

The data presented in this study are available on request from the corresponding author. The data are not publicly available due to privacy issues.

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
