# Peer review of "3D-Printed Soft Pneumatic Robotic Digit Based on Parametric Kinematic Model for Finger Action Mimicking"

_polymers, 2022, doi:10.3390/polym14142786_

Round 1

Reviewer 1 Report

Zhou et al. proposed a new modeling for the fabrication of a multi-joint pneumatic bellows actuator close to the curved contour of a human finger, and performed a comparison between the modeling and experimental results. As a material for manufacturing the multi-joint pneumatic actuator, thermoplastic polyurethane was used, and FDM 3D printing was used for the fabrication of the actuator. Thought there are no new attempts experimentally, they suggested that the new modeling improves the prediction accuracy of the moving trajectory because the start reference point is static and does not change when the soft pneumatic digit moves. However, the comparison between the previous modeling and the new modeling is only made in Fig. 6d, and it is difficult to know whether the prediction accuracy is improved in this figure. It seems necessary to mention whether the accuracy is increased through the comparison of the existing modeling and the new modeling with the experimental result in Figure 7. If the new modeling has improvements over the existing modeling in the prediction accuracy, it is recommended for the publication of the paper in Polymers. And additional comments need to be addressed is as follows.

1. In page 4, the bellow number was defined as N, but in Figure 5(a), n was used.

2 In Fig. 5(a), as N increases, the pressure to obtain bending angle decreases. It would be better if an explanation on the decrease in pressure as N increases.

3. The caption of Figure 5 is “Analysis of proposed model and experiment”. But it looks like all are modeling. Does it contain experimental results?

4. The caption of Figure 6 is the same as Figure 5. Correction is required.

5. In page 11, Figure 7(b), should be assigned to a sentence between 321 and 323.

Reviewer 2 Report

The authors propose a soft pneumatic robotic digit with pneumatic bellows actuators as a mimicked finger that combines 3D printing fabrication technique and parametric kinematic model. The theoretical model analysis and the experiment tests lead to a soft pneumatic robotic digit based on thermoplastic polyurethane where the extension and flexion performances achieve the required motions. The technical approach developed here is highly interesting and promising.

I consider this study suitable for publication only after appropriately addressing the suggestions below:

1. One missing element in this work is the discussion around basic lifting functions.

2. One minor point concerns the 3D printing fabrication: paragraph 2.3 doesn’t provide any new information besides Figure 3.b. showing the actual soft pneumatic robotic digit. This paragraph would be more suited for the Supplementary Material section.
